# Functional Effects of Permeability on Oldroyd-B Fluid under Magnetization: A Comparison of Slipping and Non-Slipping Solutions

**Muhammad Bilal Riaz** [1,2,*] , **Jan Awrejcewicz** [2,*] and **Aziz Ur Rehman** [1]

[1] Department of Mathematics, University of Management and Technology, Lahore 54770, Pakistan; s2019265005@umt.edu.pk
[2] Department of Automation, Biomechanics and Mechatronics, Lodz University of Technology, 1/15 Stefanowskiego Str., 90-924 Lodz, Poland
[*] Correspondence: bilal.riaz@umt.edu.pk (M.B.R.); jan.awrejcewicz@p.lodz.pl (J.A.)

**Abstract:** In this article, the impact of Newtonian heating in addition to slip effects was critically examined on the unsteady magnetohydrodynamic (MHD) flow of an Oldroyd-B fluid near an infinitely vertical plate. The functional effects such as the retardation and relaxation of materials can be estimated for magnetized permeability based on the relative decrease or increase during magnetization. From this perspective, a new mathematical model was formulated based on non-slippage and slippage postulates for the Oldroyd-B fluid with magnetized permeability. The heat transfer induction was also examined through a non-fractional developed mathematical model for the Oldroyd-B fluid. The exact solution expressions for non-dimensional equations of velocity and temperature were explored by employing Laplace integral transformation under slipping boundary conditions under Newtonian heating. The heat transfer rate was estimated through physical interpretation by considering the limits on the solutions induced by the Nusselt number. To comprehensively discuss the dynamics of the considered problem, the physical impacts of different parameters were studied and reverberations were graphically highlighted and deliberated. Furthermore, in order to validate the results, two limiting models, namely the Maxwell model and the second grade model, were used to compare the relevant flow characteristics. Additionally, in order to perform the parametric analysis, the graphical representation was portrayed for non-slipping and slipping solutions for velocity and temperature.

**Keywords:** magnetic effect; heat transfer; Newtonian heating; Laplace transform; porous medium; thermal radiation; physical aspect via graphs

## 1. Introduction

For a variety of fluids such as polymer solutions, certain oils, toothpaste, clay and melts, as well as blood, the specific elastic and viscous features are simultaneously those of non-Newtonian fluids. A wide range of scientists and researchers are fascinated due to its invaluable characteristics and a wide application in modern technologies and different fields of life. Non-Newtonian fluids generally differ in the three categories: namely those of rate, differential and integral types. Substantially, the precise apprehension is that all the characteristics of such fluids cannot be expressed through one unique model; this is impossible because non-Newtonian fluids have a shear rate and shear stress that are non-linearly associated with each other. The ordinary Navier–Stokes equation cannot express the different rheological features and dynamics of such fluids, e.g., their retardation, stress differences, Weissenberg effects, relaxation, elongation, micro-structure, shear thickening/thinning, memory effects, re-coil, and yield stress. To conveniently handle the involved nonlinear additional term and to anticipate the rheological characteristics of non-Newtonian fluids, a large variety of different models have been proposed such

as Jeffery's model [1], the Maxwell model [2], the Burgers' viscoelastic model [3], second grade fluid model [4], Sisko's model [5] and the Oldroyd-B model [6]. In recent years, the Oldroyd-B fluid, among the many rate types of fluids, has gained a unique status because the classical Newtonian fluid and the Maxwell fluid are special cases of the Oldroyd-B fluid. The Oldroyd-B model exhibits the relaxation and retardation mechanisms and it is a very simple model which appropriately expresses the elastic and viscous behaviors of the fluid because the Oldroyd-B fluid has potential which also includes the flow history. The velocity field and stress field for the Oldroyd-B fluid were analytically examined for a constantly moving plate by Fetecau et al. [7]. Fetecau et al. [8] also extended the same investigation to explore transient channel transport for the Oldroyd-B model which settled due to the instinctive motions of the plate. Gul et al. [9] investigated the thin film motion for the transient MHD Oldroyd-B model over an oscillating belt. Tiwana et al. [10] studied the influence of ramped boundary velocity, ramped wall heating and a permeable medium on the convective transport of transient MHD Oldroyd-B fluid. Recently, the Littlewood-Paley theory was used by Wan [11] to scrutinize the universal well-posed particularities of Oldroyed-B fluid two-dimensional flow under initial conditions. The transient motion of the Oldroyd-B fluid with the existence of influential cohesion forces and its flow induced by the translational movement of the surface were inspected by Shakeel et al. [12]. Tahir et al. [13] investigated the solution in series form to inspect the fractional behavior of an Oldroyd-B fluid flow for two revolving cylinders. The same investigation was extended by Wang et al. [14], who applied an integral transform method to express the derived results' modified Bessel function. Elhanafy et al. [15] used the numerical solution of the Oldroyd-B model to determine the blood's movement across the abdominal aortic section. The finite difference method was applied inside a straight cloture with expanding boundaries to manipulate the heat transfer and MHD motion in an Oldroyd-B model by Ali et al. [16]. It is well-accepted fact that heat transfer from a high to low temperature wall occurs through the certain movement of a fluid—such a mechanism is called "heat convection" (advection). This is of great importance as fluid motion occurs on the basis of natural or forced convection [17–22]. Solangi et al. [23] discussed the heat transfer characteristics of large concentrations. The key point highlighted in this work is that the heat and mass behavior of the fluid are controlled by the particle's size. Shafiq et al. [24] conducted a stimulating study for MHD convective flow with the parametric analysis of the proposed problem for thermophoretic, Blackian motion, buoyancy forces, Newtonian heating and magnetic field for concentration and temperature. Two solutions which have numerical stability were used to examine the dual results of the governing partial differential equation investigated by Hamid et al. [25]. Abdelmalek et al. [26] used the control volume finite element scheme, also known as hybrid technique, for the curved circular shape heater in addition to nanoparticles for heat transmission. The thermo-diffusion effects on the time-dependent free convective fluid flow of applying surface modification technology were determined by Kashif [27]. Further relevant studies regarding heat and mass have been studied in detail by: heat transfer approaches via analytical [28–35], numerical [36–41], with the application of fractional operators [42–45] and multi-dimensional [46–51] types.

To effectively study the dynamics of fluid flow problems, the no-slip and slip conditions were generally assumed at the boundary. In the case of the no-slip conditions, it is supposed that there is no relative movement of the fluid at the boundary, i.e., the fluid and the boundary are stationary and the speed of the flow is zero. Physically, between the surface of the fluid and the boundary, the adherence phenomenon is more dominant than the cohesive phenomenon. The no-slip condition has great applications in different practical use, regardless of a few coupled compulsions, because the complexity of flow dynamics are reduced in the case of no-slip conditions. However, some smooth surfaces exist where cohesion forces show dominance and in this case, a relative movement in the fluid was observed before the fluid moved from the boundary. For such a type of surfaces, accurate results in the case of no-slip conditions cannot produce fluid flow properties. For example,

the analysis of blood transportation through arteries cannot be studied using no-slip conditions [52]. Navier presented a new technique to effectively handle such problems by using slip conditions [53]. The slip condition is strongly applicable in different areas of daily life, for example, industrial lubricants, soil degradation by erosion, medical fields—especially cleaning the artificial heart valves, by applying protrusion, different biological fluids and various type of nanofluids in porous media [54]. Related investigations on the subject of slip conditions are discussed in [55].

The objective of this study was to analyze the impact of Newtonian heating in addition to slip effects by critically examining the unsteady MHD flow of an Oldroyd-B fluid near an infinitely vertical plate. Furthermore, the functional effects such as the retardation and relaxation of materials can be estimated for magnetized permeability based on relative decrease or increase during magnetization. Exact solution expressions for non-dimensional equations of velocity and temperature were explored by employing Laplace integral transformation under slipping boundary conditions under Newtonian heating. Moreover, the heat transfer rate was estimated through physical interpretation by considering the limits on the solutions induced by the Nusselt number. The physical impacts of different parameters were studied and the reverberations were graphically highlighted and deliberated. Furthermore, two limiting models, namely the Maxwell model and second grade model, were used to compare the relevant flow characteristics in order to validate the results. Finally, in order to perform the parametric analysis, the graphical representation was portrayed for non-slipping and slipping solutions for velocity and temperature.

## 2. Mathematical Model

Consider the unsteady magnetohydrodynamic (MHD) flow of an Oldroyd-B fluid near an infinitely long plate with heat transfer under Newtonian heating. Suppose that the external magnetic forces act along the normal direction of the movement of the fluid and that the fluid is also electrically conducted. The impact of the thermal radiation is parallel to the plate but is assumed to be insignificant—the opposite is true in the horizontal direction. Velocity is considered at any point that does not only depend upon the radial distance in the proposed problem but also on the horizontal direction, i.e., the x distance. Thus, the flow in this problem is a two-dimensional flow. First of all, it is presumed that a system with no movement is a system under rest conditions. After a short interval of time, the fluid starts to move due to mixed convection, and the fluid flows along the plate as illustrated in Figure 1.

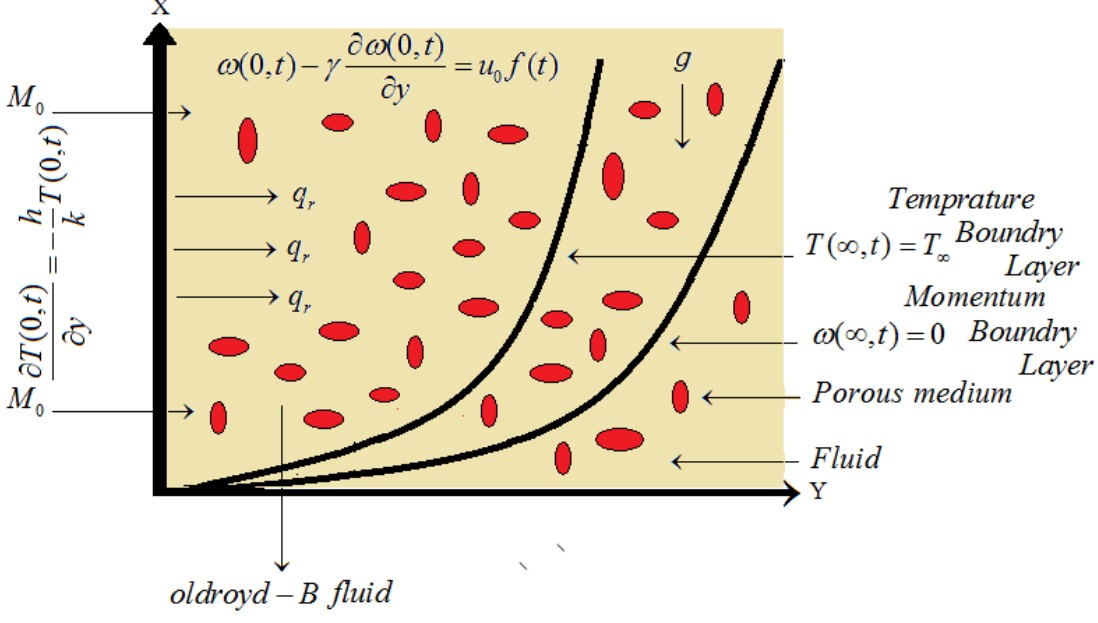

**Figure 1.** Geometry of the considered model.

The configured flow equations for an Oldroyd-B fluid model are given by [56,57]

$$\nabla.V = 0,\tag{1}$$

$$\nabla.T + \rho g + J \times M = \rho\left[(V.\nabla)V + \frac{\partial V}{\partial t}\right].\tag{2}$$

where $V$, $J$, $T$, $M$, $\rho$, $\rho g$ and $t$ are the parameters used in the above equations to denote the velocity field, electric density, Cauchy stress tensor, total magnetic field (having both induced and imposed forces), fluid density, body force and time, respectively. The relation between the Cauchy stress and extra stress tensor is represented by $T$ and $S$, respectively, for the Oldroyd-B fluid formulated as

$$S = pI + T,\tag{3}$$

$$\left(S + \lambda_1\frac{\partial S}{\partial t}\right) = \mu\left(1 + \lambda_2\frac{\partial}{\partial t}\right)A_1,\tag{4}$$

$$A_1 = (\nabla V)^T + (\nabla V),\tag{5}$$

where $\lambda_1$, $p$, $I$, $\mu$, $\lambda_2$, $-pI$, $\frac{D}{Dt}$ and $A_1$ denote the relaxation time, pressure, identity tensor, dynamic viscosity, retardation time, tensor's indeterminate part, convective time derivative and Rivlin–Ericksen tensor, respectively. Furthermore, the Maxwell equations for the electric and magnetic fields are written in the following form:

$$\Delta.M = 0, \quad \Delta \times E = -\frac{\partial M}{\partial t}, \quad \Delta \times M = \mu_m J, \quad J \times M = -\sigma M_0^2 V.\tag{6}$$

where $J$ represents the electric field, $M$ is denotes the magnetic field, $\sigma$ represents the fluid's electrical conductivity and $\mu_m$ denotes the magnetic permeability. Furthermore, $M = M_0 + M_1$, in which $M_0$ and $M_1$ denote the imposed and induced magnetic fields, but the induced magnetic field is not considered herein. In the present work, the velocity field is considered in the following form:

$$V = V(y, t) = w(y, t)i \quad and \quad S = S(y, t).\tag{7}$$

where $i$ and $w$ represent the unit vector in the x direction and the x component of velocity $V$, respectively. By substituting Equations (3)–(7) into Equation (2) and after simplification via the application of the Rosseland approximation and Boussinesq's approximation, we obtained the principal governing equation for the MHD Oldroyd-B fluid in addition to the initial/boundary conditions which are given as [58,59]

$$\left(1 + \lambda_1\frac{\partial}{\partial t}\right)\frac{\partial w(y, t)}{\partial t} = v\left(1 + \lambda_2\frac{\partial}{\partial t}\right)\frac{\partial^2 w(y, t)}{\partial y^2} + g\beta\left(1 + \lambda_1\frac{\partial}{\partial t}\right)(T(y, t) - T_\infty)$$
$$- \left(1 + \lambda_1\frac{\partial}{\partial t}\right)\frac{\sigma M_0^2}{\rho}w(y, t),\tag{8}$$

$$C_p\frac{\partial T(y, t)}{\partial t} = \frac{k}{\rho}\frac{\partial^2 T(y, t)}{\partial y^2},\tag{9}$$

$$\left(1 + \lambda_1\frac{\partial}{\partial t}\right)S = \mu\frac{\partial w(y, t)}{\partial y},\tag{10}$$

with the initial and boundary conditions:

$$T(y,0) = T_\infty, \quad w(y,0) = 0, \quad \frac{\partial w(y,0)}{\partial y} = 0, \quad \frac{\partial w(y,0)}{\partial t} = 0, \quad y \geq 0, \tag{11}$$

$$w(0,t) - \gamma \frac{\partial w(0,t)}{\partial y} = u_0 f(t), \quad \frac{\partial T(0,t)}{\partial y} = -\frac{h}{k} T(0,t), \quad t \geq 0, \tag{12}$$

$$w(y,t) \to 0, \quad T(y,t) \to \infty \quad as \quad y \to \infty. \tag{13}$$

To non-dimensionalize the governing partial differential equation, the following set of variables was introduced:

$$t^* = \frac{vh^2}{k^2} t, \quad y^* = \frac{h}{k} y, \quad w^* = \frac{w}{u_0}, \quad \theta = \frac{T - T_\infty}{T_\infty}, \quad u_0^2 = \frac{v^2 h^2}{k^2}, \quad Gr = \frac{g \beta v T_\infty}{u_0^3},$$

$$M = \frac{k^2 \sigma M_0^2}{h^2 \mu}, \quad \lambda_1^* = \frac{vh^2}{k^2} \lambda_1, \quad \lambda_2^* = \frac{vh^2}{k^2} \lambda_2, \quad Pr = \frac{vC_p}{k}, \quad \gamma^* = \frac{h}{k} \gamma, \quad S^* = \frac{k}{h} \frac{S}{u_0 \mu}, \tag{14}$$

The partial differential equations in dimensionless form, after removing the asterisk $*$ notation, are written as

$$\left(1 + \lambda_1 \frac{\partial}{\partial t}\right) \frac{\partial w(y,t)}{\partial t} = \left(1 + \lambda_2 \frac{\partial}{\partial t}\right) \frac{\partial^2 w(y,t)}{\partial y^2} + \left(1 + \lambda_1 \frac{\partial}{\partial t}\right) Gr\theta - \left(1 + \lambda_1 \frac{\partial}{\partial t}\right) Mw(y,t), \tag{15}$$

$$\frac{\partial \theta(y,t)}{\partial t} = \frac{1}{Pr} \frac{\partial^2 \theta(y,t)}{\partial y^2}, \tag{16}$$

$$\left(1 + \lambda_1 \frac{\partial}{\partial t}\right) S = \frac{\partial w(y,t)}{\partial y}. \tag{17}$$

and the corresponding set of initial and boundary conditions are stated as

$$w(y, \chi_0) = 0, \quad \theta(y, \chi_0) = 0, \quad w_t(y, \chi_0) = 0, \quad w_y(y, \chi_0) = 0, \tag{18}$$

$$w(\chi_0, t) - \gamma \frac{\partial w(\chi_0, t)}{\partial y} = f(t), \quad \frac{\partial \theta(\chi_0, t)}{\partial y} = -(1 + \theta(\chi_0, t)), \tag{19}$$

$$w(y,t) \to 0, \quad \theta(y,t) \to 0, \quad as \quad y \to \infty, \quad t \geq 0, \quad \chi_0 = 0. \tag{20}$$

### 3. Solution of the Problem

*3.1. Exact Solution of Heat Profile*

To obtain the solution by employing the Laplace integral transformation on Equation (16) by applying the conditions, we have:

$$\frac{\partial^2 \bar{\theta}(y,s)}{\partial y^2} - sPr\bar{\theta}(y,s) = 0. \tag{21}$$

The solution for Equation (21) is obtained and written as:

$$\bar{\theta}(y,s) = c_1 e^{y\sqrt{Prs}} + c_2 e^{-y\sqrt{Prs}}. \tag{22}$$

To determine the constants $c_1$ and $c_2$ involved in using the stated conditions for the applied temperatures, we have:

$$\bar{\theta}(y,s) = \frac{e^{-y\sqrt{Prs}}}{s\left(\sqrt{Prs} - 1\right)}, \tag{23}$$

$$\bar{\theta}(y,s) = -\frac{\xi e^{-q\sqrt{s}}}{s(\sqrt{s}+\xi)}. \tag{24}$$

The inverse Laplace integral transformation employed herein to obtain the required solution of Equation (24) is written as

$$\theta(y,t) = -\left[e^{-\xi q}e^{\xi^2 t}erfc\left(\xi\sqrt{t}+\frac{q}{2\sqrt{t}}\right)+erfc\left(\frac{q}{2\sqrt{t}}\right)\right]. \tag{25}$$

Nusselt Number

To estimate the heat transfer rate, the Nusselt number is calculated as

$$
\begin{aligned}
N_u &= -\frac{\partial\theta(y,t)}{\partial y}|_{y=0},\\
&= -\frac{\partial}{\partial y}\mathcal{L}^{-1}\{\bar{\theta}(y,s)\}|_{y=0},\\
&= -\mathcal{L}^{-1}\left\{\frac{\partial\bar{\theta}(y,s)}{\partial y}|_{y=0}\right\},\\
&= \mathcal{L}^{-1}\left\{\frac{\sqrt{Prs}}{s(\sqrt{Prs}-1)}\right\},\\
&= \mathcal{L}^{-1}\left\{\frac{1}{\sqrt{s}(\sqrt{s}+\xi)}\right\},\\
&= e^{\xi^2 t}erfc(\xi\sqrt{t}).
\end{aligned}
\tag{26}
$$

where $\quad \xi = -\frac{1}{\sqrt{Pr}}$ and $q = y\sqrt{Pr}$.

### 3.2. Exact Solution of Velocity Profile

To obtain the solution by employing the Laplace integral transformation on Equation (15) by applying conditions, we have:

$$s\bar{w}(y,s) + \lambda_1 s^2\bar{w}(y,s) = (1+\lambda_2 s)\frac{\partial^2\bar{w}(y,s)}{\partial y^2} + (1+\lambda_1 s)G_r\bar{\theta}(y,s) - (1+\lambda_1 s)M\bar{w}(y,s). \tag{27}$$

by using the Equation (23) for the value $\bar{\theta}(y,s)$, Equation (27) has solution in the form:

$$\bar{w}(y,s) = c_3 e^{y\sqrt{\frac{\lambda_1 s^2+as+M}{1+\lambda_2 s}}} + c_4 e^{-y\sqrt{\frac{\lambda_1 s^2+as+M}{1+\lambda_2 s}}} - \left(\frac{Gr(1+\lambda_1 s)e^{-y\sqrt{Prs}}}{s(\sqrt{Prs}-1)(Prs(1+\lambda_2 s)-(\lambda_1 s^2+as+M))}\right). \tag{28}$$

After substituting the values of the constants $c_3$ and $c_4$ in Equation (28), velocity expressions are written as

$$
\begin{aligned}
\bar{w}(y,s) &= \frac{F(s)e^{-y\sqrt{\frac{\lambda_1 s^2+as+M}{1+\lambda_2 s}}}}{1+\gamma\sqrt{\frac{\lambda_1 s^2+as+M}{1+\lambda_2 s}}} - \left(\frac{Gr(1+\lambda_1 s)e^{-y\sqrt{Prs}}}{s(\sqrt{Prs}-1)(Prs(1+\lambda_2 s)-(\lambda_1 s^2+as+M))}\right)\times\\
&\quad\left(\frac{(1+\gamma\sqrt{Prs})e^{-y\sqrt{\frac{\lambda_1 s^2+as+M}{1+\lambda_2 s}}}}{1+\gamma\sqrt{\frac{\lambda_1 s^2+as+M}{1+\lambda_2 s}}} - e^{-y\sqrt{Prs}}\right).
\end{aligned}
\tag{29}
$$

where $a = 1 + \lambda_1 M$.

Equation (29) can be rearranged in another form as

$$\bar{w}(y,s) = F(s)\bar{A}_0(y,s) - \frac{\xi Gr}{a_1}\frac{1}{(\sqrt{s}+\xi)}\left[\frac{a_5}{s} + \frac{a_6}{s+m_1} + \frac{a_7}{s+n_1}\right][\bar{A}_0(y,s) + \bar{w}_4(y,s) - \bar{w}_5(y,s)], \tag{30}$$

$$\bar{w}(y,s) = F(s)\bar{A}_0(y,s) - \frac{\xi Gr}{a_1}[a_5\bar{w}_1(y,s) + a_6\bar{w}_2(y,s) + a_7\bar{w}_3(y,s)][\bar{A}_0(y,s) + \bar{w}_4(y,s) - \bar{w}_5(y,s)], \tag{31}$$

where:

$$\bar{A}_0(y,s) = e^{-y\sqrt{\frac{\lambda_1 s^2 + as + M}{1+\lambda_2 s}}} \cdot \frac{1}{1 + \gamma\sqrt{\frac{\lambda_1 s^2 + as + M}{1+\lambda_2 s}}},$$

$$= \left[\sum_{\alpha=0}^{\infty}\frac{(-y)^{\alpha}}{\alpha!}\left(\frac{\lambda_1 s^2 + as + M}{1+\lambda_2 s}\right)^{\frac{\alpha}{2}}\right]\left[\sum_{\beta=0}^{\infty}(-1)^{\beta}(\gamma)^{\beta}\left(\frac{\lambda_1 s^2 + as + M}{1+\lambda_2 s}\right)^{\frac{\beta}{2}}\right],$$

By applying the discrete convolution known as the Cauchy product, each of which have m terms with two truncated series, yields:

$$\bar{A}_0(y,s) = \sum_{\alpha=0}^{m}\sum_{\beta=0}^{m}\frac{(-y)^{\alpha}(-1)^{m-\beta}(\gamma)^{m-\beta}}{\alpha!}\left(\frac{\lambda_1 s^2 + as + M}{1+\lambda_2 s}\right)^{m+\frac{\alpha}{2}-\frac{\beta}{2}},$$

$$= \sum_{\alpha=0}^{m}\sum_{\beta=0}^{m}\sum_{\eta=0}^{\infty}\sum_{l=0}^{\infty}\sum_{k=0}^{\infty}\frac{(-y)^{\alpha}(-1)^{m-\beta}(\gamma)^{m-\beta}(c)^{m-\eta+\frac{\alpha}{2}-\frac{\beta}{2}}(b)^{\eta-l}(d)^l(\lambda_2)^k\Gamma(l+k)\Gamma(\eta+1)}{(\alpha!)(\eta!)(l!)(k!)\Gamma(l)\Gamma(\eta-l+1)} \cdot$$

$$\frac{\Gamma(m+\frac{\alpha}{2}-\frac{\beta}{2}+1)}{\Gamma(m+\frac{\alpha}{2}-\frac{\beta}{2}-\eta+1)}\cdot\frac{1}{s^{l-\eta-k}},$$

$$A_0(y,t) = \sum_{\alpha=0}^{m}\sum_{\beta=0}^{m}\sum_{\eta=0}^{\infty}\sum_{l=0}^{\infty}\sum_{k=0}^{\infty}\frac{(-y)^{\alpha}(-1)^{m-\beta}(\gamma)^{m-\beta}(c)^{m-\eta+\frac{\alpha}{2}-\frac{\beta}{2}}(b)^{\eta-l}(d)^l(\lambda_2)^k\Gamma(l+k)\Gamma(\eta+1)}{(\alpha!)(\eta!)(l!)(k!)\Gamma(l)\Gamma(\eta-l+1)} \cdot$$

$$\frac{\Gamma(m+\frac{\alpha}{2}-\frac{\beta}{2}+1)}{\Gamma(m+\frac{\alpha}{2}-\frac{\beta}{2}-\eta+1)}\cdot\frac{t^{l-\eta-k-1}}{\Gamma(l-\eta-k)},$$

$$a_0 = a - \frac{\lambda_1}{\lambda_2}, \quad b = \frac{\lambda_1}{\lambda_2}, \quad c = \frac{a_0}{\lambda_2}, \quad d = M - \frac{a_0}{\lambda_2},$$

$$a_1 = Pr\lambda_2 - \lambda_1, \quad a_2 = \frac{Pr-a}{a_1}, \quad a_3 = \frac{M}{a_1}, \quad a_4 = \sqrt{a_3 + \frac{a_2^2}{4}},$$

$$m_1 = a_2 - a_4, \quad n_1 = a_2 + a_4, \quad \xi = -\frac{1}{\sqrt{Pr}}, \quad a_5 = \frac{1}{m_1 n_1},$$

$$a_6 = \frac{\lambda_1 m_1 - 1}{m_1(n_1 - m_1)}, \quad a_7 = \frac{\lambda_1 n_1 - 1}{n_1(m_1 - n_1)},$$

$$\mathcal{L}^{-1}(\bar{w}_1(y,s)) = w_1(y,t) = (f*f_1)(t),$$

$$\mathcal{L}^{-1}(\bar{w}_1(y,s)) = w_2(y,t) = (f*f_2)(t),$$

$$\mathcal{L}^{-1}(\bar{w}_1(y,s)) = w_3(y,t) = (f*f_3)(t),$$

$$\mathcal{L}^{-1}(\bar{w}_4(y,s)) = w_4(y,t) = (f_4*A_0)(t),$$

$$\mathcal{L}^{-1}(e^{-q\sqrt{s}}) = w_5(y,t) = \frac{q}{2\sqrt{\pi t^3}}e^{-\frac{1}{4t}q^2} \quad as \quad q = y\sqrt{Pr},$$

$$f(t) = \mathcal{L}^{-1}\left(\frac{1}{\sqrt{s}+\xi}\right) = \frac{1}{\sqrt{\pi t}} - \xi e^{\xi^2 t} erfc(\xi\sqrt{t}), \quad f_1(t) = \mathcal{L}^{-1}\left(\frac{1}{s}\right) = 1,$$

$$f_2(t) = \mathcal{L}^{-1}\left(\frac{1}{s+m_1}\right) = e^{-m_1 t}, \quad f_3(t) = \mathcal{L}^{-1}\left(\frac{1}{s+n_1}\right) = e^{-n_1 t},$$

$$f_4(t) = \mathcal{L}^{-1}\left(\gamma\sqrt{Pr}\sqrt{s}\right) = -\frac{\gamma\sqrt{Pr}}{2\sqrt{\pi}t^{\frac{3}{2}}} \tag{32}$$

Finally, the solution which requires employing the inverse Laplace transformation for the momentum equation with the convolution product on Equation (31) has the following form:

$$
\begin{aligned}
w(y,t) = & f(t) * A - \frac{a_5}{a_1}\xi Gr(w_1 * A) - \frac{a_6}{a_1}\xi Gr(w_2 * A) - \frac{a_7}{a_1}\xi Gr(w_3 * A) \\
& - \frac{a_5}{a_1}\xi Gr(w_1 * w_4) - \frac{a_6}{a_1}\xi Gr(w_2 * w_4) - \frac{a_7}{a_1}\xi Gr(w_3 * w_4) \\
& + \frac{a_5}{a_1}\xi Gr(w_1 * w_5) + \frac{a_6}{a_1}\xi Gr(w_2 * w_5) + \frac{a_7}{a_1}\xi Gr(w_3 * w_5).
\end{aligned}
\tag{33}
$$

We recover the Maxwell model by considering time retardation parameter of value zero, i.e., $\lambda_2 = 0$, as acquired by Ghalib et al. [59]. Furthermore, the velocity field solution for the second grade fluid is traced out by considering $\lambda_1 = 0$ in Equation (33), as derived by Aziz et al. [34], which validates our current results with the previous literature.

## 4. Results and Discussion

In the present work, the flow of the Oldroyd-B fluid was investigated, and the exact expressions of the analytical solutions to non-dimensional equations of temperature and velocity were explored by the application of the Laplace integral transformation with the slipping boundary conditions under Newtonian heating. For many reasons, these exact analytical solutions for dimensionless velocity and temperature are very important. For instance, in order to explore the accuracies of many computed approximate solutions by using numerical techniques, for complex flow phenomena and these solutions in various fields of applied sciences and engineering which have great importance, exact analytical results are mandatory. Thus, closed-form solutions are essential to describe the non-Newtonian fluids' behavior. Many graphs have been portrayed to examine the effects of various physical parameters $\lambda_1$, $M$, $Pr$, $Gr$ and $\lambda_2$. The graphical demonstration for the temperature profile and velocity field were generated to correspond to several connected parameters by using the Mathcad software. For the velocity field solution, all diagrams are plotted corresponding to the slip and no-slip boundary conditions for the Oldroyd-B fluid model.

In Figure 2, the effects of the Prandtl number for the temperature profile are displayed. It can be easily deduced from these graphs that the temperature profile decreases in function of the increasing $Pr$. Generally, the thickness of the thermal outline layer rapidly decreases as the values of $Pr$ linearly increase due to the deceleration of this temperature curve.

In Figures 3–7, the graphs for the velocity field for the slip and no-slip boundary conditions are portrayed for the function $f(t) = e^{at}$ by taking the values $a = 0.25$ and $t = 1.5$ for all diagrams. Moreover, Figures 8–12 illustrate the graphical behavior of the velocity field for the slip and no-slip boundary conditions for the function $f(t) = sint$.

In Figures 3 and 8, the effects of the Prandtl number $Pr$ on the velocity field under slip conditions and the corresponding velocity profile under no-slip conditions are shown. It can be noted in these graphs that both the velocity under slip conditions as well as the velocity under no-slip conditions decline with the advancement of the Prandtl number.

In Figures 4 and 9, the graphs for the velocity field under slip conditions and the corresponding velocity profile under no-slip conditions are illustrated for the related role of viscosity and the buoyancy forces in the movement of the fluid are depicted. It can be observed that the velocity profile escalated in function of the enhancement of the values

of *Gr* for both cases. When the positive values of *Gr* are considered, this causes the fluid temperature to increase, i.e., by turning into free convection currents in the movement of the fluid region. A strong buoyancy force is produced in the flowing region relative to the increasing values of *Gr*. Due to this strong buoyancy force, all the viscous forces become powerless, leaving one to appreciate the fluid velocity.

In Figures 5 and 10, the impacts of the magnetic field *M* on the velocity field under slip conditions and the corresponding velocity profile under no-slip conditions are illustrated. It can be easily perceived from these graphs that both the velocity under slip conditions as well as the velocity under no-slip conditions decelerate in function of the increase in the strength of the magnetic field. Resistive type forces are termed Lorentz forces and these are generated due to the imposition of a magnetic field. These forces behave similarly to dragging forces which suppress the forces that help the fluid flow. Consequently, retardation in the fluid flow ultimately causes the deceleration in the motion of the fluid and in due course the fluid comes to a halt.

In Figures 6 and 11, the graphs for the velocity under slip conditions and the corresponding velocity profile under no-slip conditions are demonstrated to analyze the relaxation parameter $\lambda_1$. Viscous forces become weaker as the values of $\lambda_1$ increase. Thus, it can be seen from the graphs that for both cases, the velocity profile accelerates in function of the large values of $\lambda_1$.

In Figures 7 and 12, the influence of the retardation parameter $\lambda_2$ on the velocity field under slip conditions and in relation to the velocity profile under no-slip conditions are represented. The retardation parameter and velocity share are inversely related. It can be easily seen that both the velocity under slip conditions as well as the velocity under no-slip conditions decelerate in function of the increase in the values of $\lambda_2$.

From all graphs, it can be noted that similar curve trends are observed for fluid flow under slip conditions and under no-slip conditions. Furthermore, it was analyzed that for both functions $f(t) = e^{at}$ and $f(t) = sint$, the velocity field presents the same curve pattern for all involved system parameters.

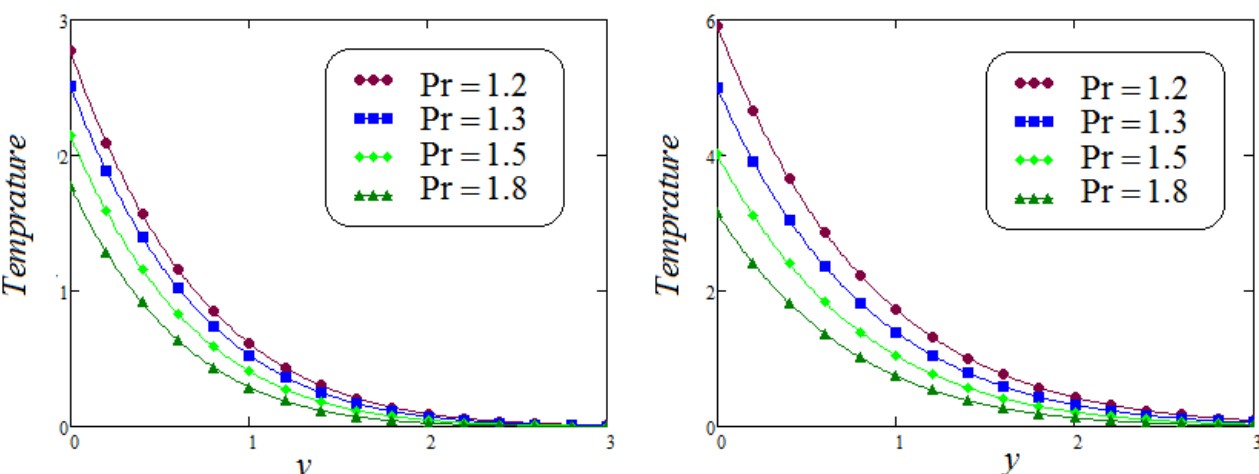

**Figure 2.** Temperature profile using different *Pr* values, when $t = 0.9$ and $t = 1.5$.

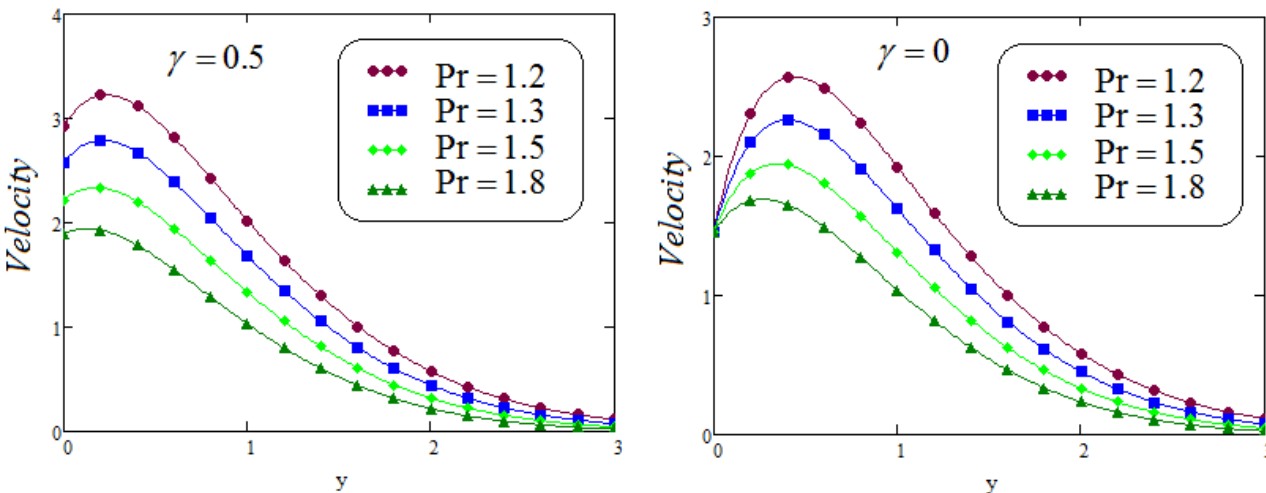

**Figure 3.** Velocity field under slip conditions as well as the velocity field under no-slip conditions for different *Pr* values when $f(t) = e^{at}$, $a = 0.25$, $t = 1.5$, $Gr = 3.5$, $\lambda_1 = 0.6$, $\lambda_2 = 0.2$, $M = 2$.

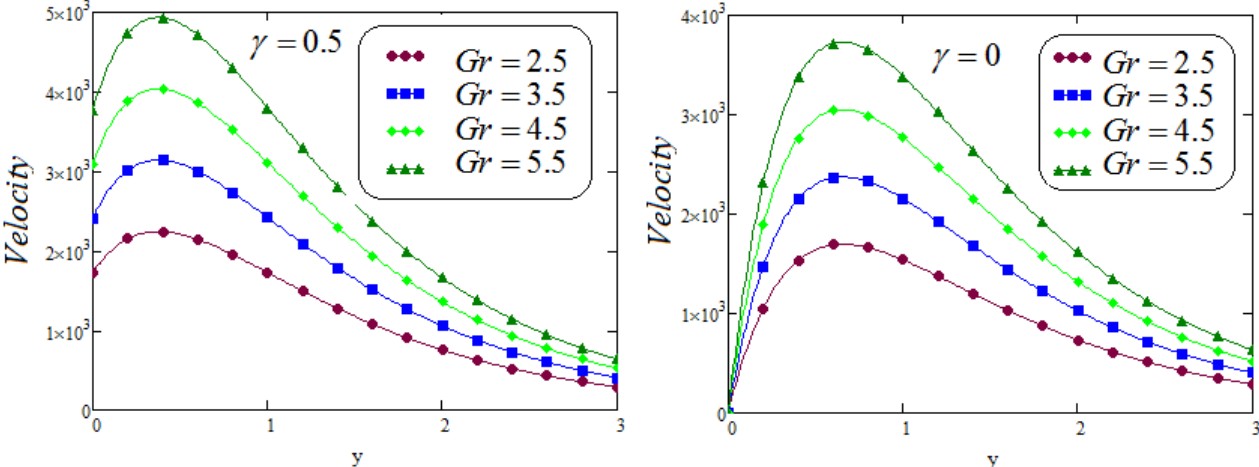

**Figure 4.** Velocity field under slip conditions as well as the velocity field under no-slip conditions for different *Gr* values when $f(t) = e^{at}$, $a = 0.25$, $t = 1.5$, $Pr = 0.71$, $\lambda_1 = 0.6$, $\lambda_2 = 0.2$, $M = 2$.

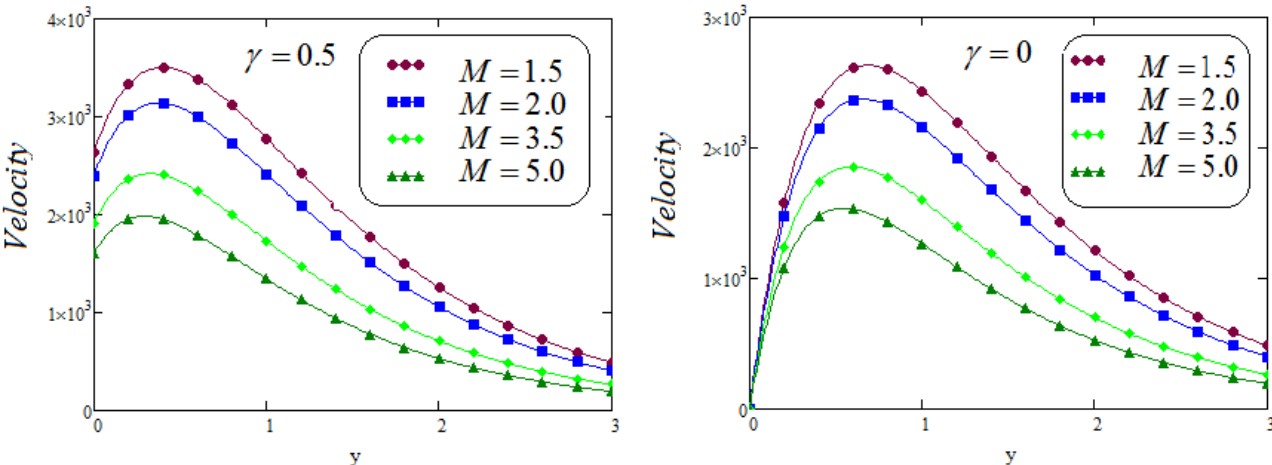

**Figure 5.** Velocity field under slip conditions as well as the velocity field under no-slip conditions for different *M* values when $f(t) = e^{at}$, $a = 0.25$, $t = 1.5$, $Gr = 3.5$, $\lambda_1 = 0.6$, $\lambda_2 = 0.2$, $Pr = 0.71$.

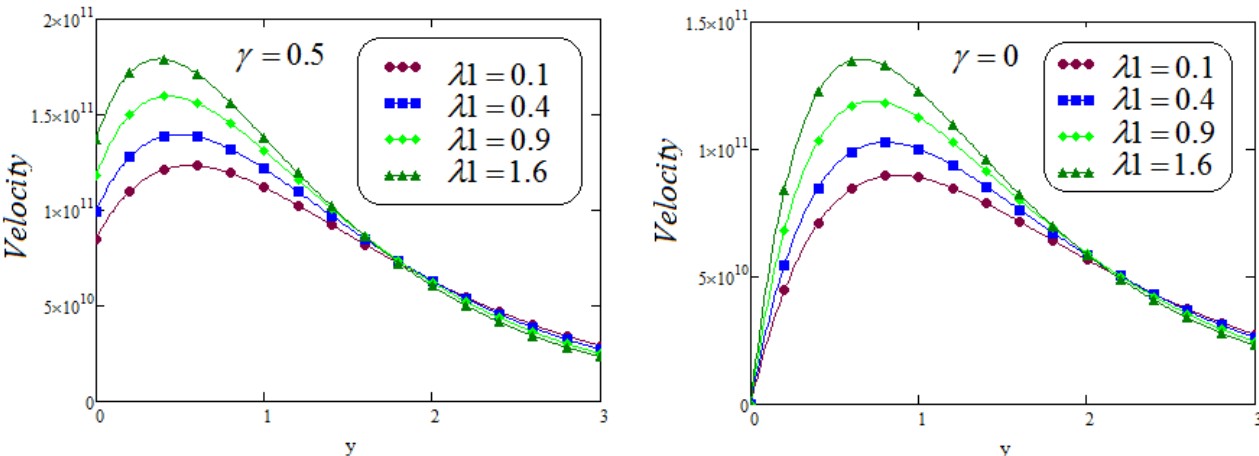

**Figure 6.** Velocity field under slip conditions as well as the velocity field under no-slip conditions for different $\lambda_1$ values when $f(t) = e^{at}$, $a = 0.25$, $t = 1.5$, $Gr = 3.5$, $Pr = 0.71$, $\lambda_2 = 0.2$, $M = 2$.

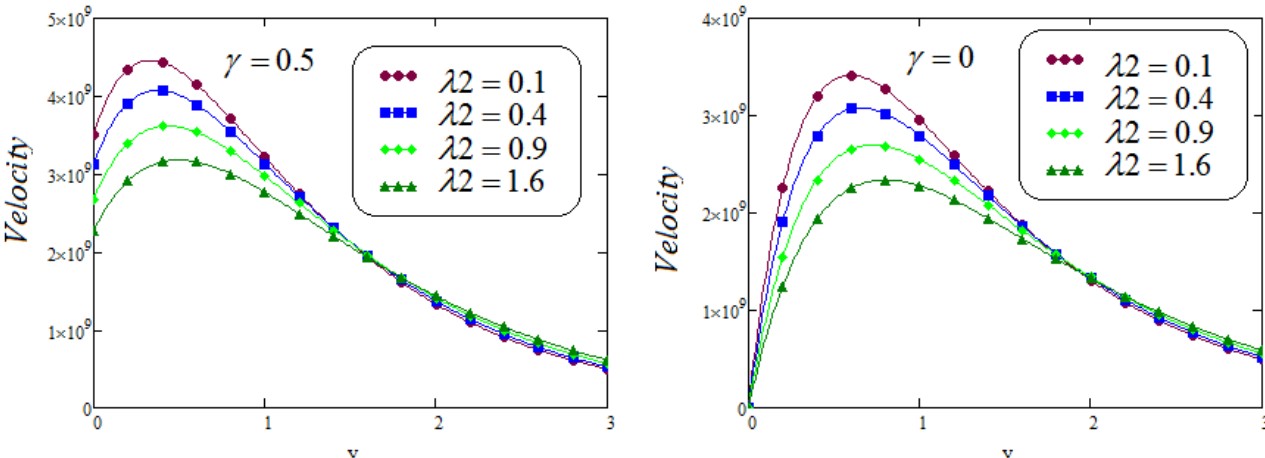

**Figure 7.** Velocity field under slip conditions as well as the velocity field under no-slip conditions for different $\lambda_2$ values when $f(t) = e^{at}$, $a = 0.25$, $t = 1.5$, $Gr = 3.5$, $\lambda_1 = 0.6$, $Pr = 0.71$, $M = 2$.

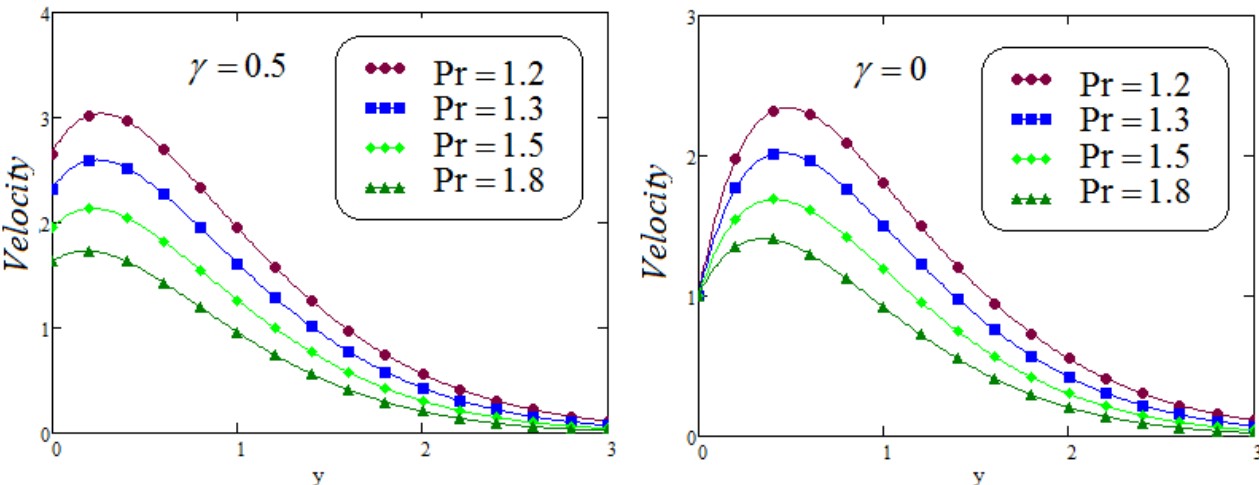

**Figure 8.** Velocity field under slip conditions as well as the velocity field under no-slip conditions for different $Pr$ values when $f(t) = sint$, $t = 1.5$, $Gr = 3.5$, $\lambda_1 = 0.6$, $\lambda_2 = 0.2$, $M = 2$.

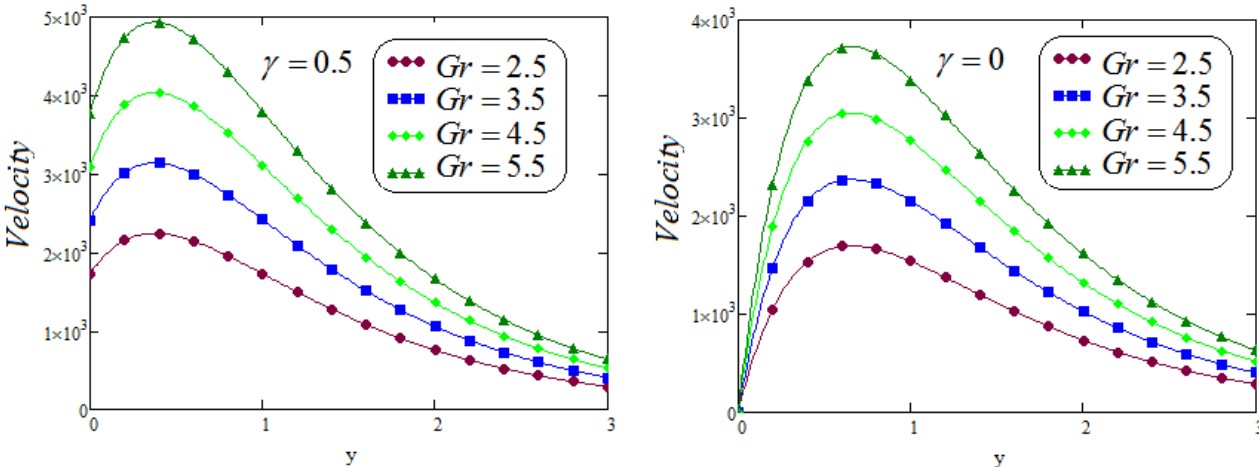

**Figure 9.** Velocity field under slip conditions as well as the velocity field under no-slip conditions for different *Gr* values when $f(t) = sint$, $t = 1.5$, $Pr = 0.71$, $\lambda_1 = 0.6$, $\lambda_2 = 0.2$, $M = 2$.

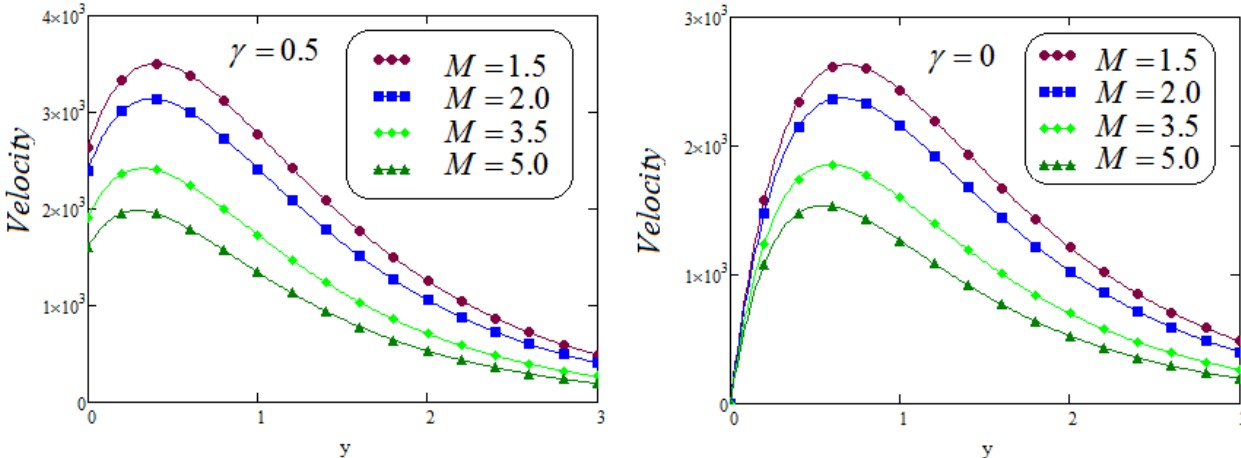

**Figure 10.** Velocity field under slip conditions as well as the velocity field under no-slip conditions for different *M* values when $f(t) = sint$, $t = 1.5$, $Gr = 3.5$, $\lambda_1 = 0.6$, $\lambda_2 = 0.2$, $Pr = 0.71$.

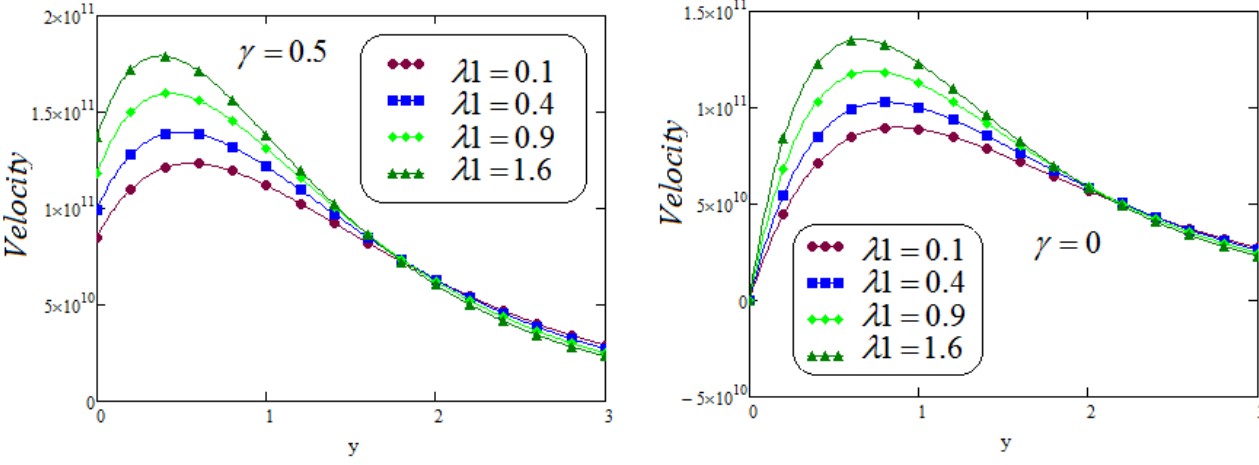

**Figure 11.** Velocity field under slip conditions as well as the velocity field under no-slip conditions for different $\lambda_1$ values when $f(t) = sint$, $t = 1.5$, $Gr = 3.5$, $Pr = 0.71$, $\lambda_2 = 0.2$, $M = 2$.

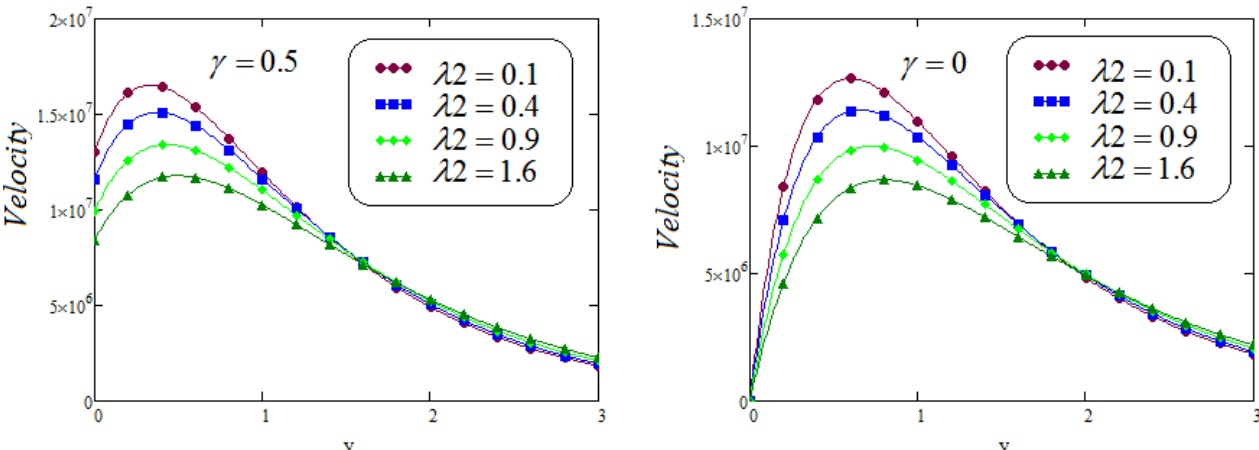

**Figure 12.** Velocity field under slip conditions as well as the velocity field under no-slip conditions for different $\lambda_2$ values when $f(t) = sint$, $t = 1.5$, $Gr = 3.5$, $\lambda_1 = 0.6$, $Pr = 0.71$, $M = 2$.

## 5. Conclusions

A thorough investigation of Newtonian heating in addition to slip effects on the unsteady magnetohydrodynamic (MHD) flow of an Oldroyd-B fluid near an infinitely long plate was analyzed in this research. The exact expressions of the analytical solution to non-dimensional equations of velocity and temperature were explored by employing the Laplace integral transformation under slipping boundary conditions under Newtonian heating. Many graphs were portrayed to examine the effects of various physical parameters such as the time relaxation parameter, $\lambda_1$, magnetic number $M$, Prandtl number $Pr$, Grashof number $Gr$ and the time retardation parameter $\lambda_2$. The graphical demonstration for the temperature profile and velocity field for several connected parameters under slip conditions and under no-slip conditions. The obtained results are summarized as:

- The temperature graphs show that the temperature profile decreases for higher values of $Pr$.
- The graphs for the velocity field under slip conditions as well as the velocity field under no-slip conditions show that the effects of $\lambda_1$ and $\lambda_2$ on the velocity contour are quite the opposite.
- From the graphs, one can see that the elevated values of $M$ and $Pr$ reduced the velocity curve.
- The velocity profile was stimulated in function of the increasing values of $Gr$.
- It can be observed that the velocity profile for no-slip flow is lower than the velocity profile for slip flow.
- It was analyzed that for both functions $f(t) = e^{at}$ and $f(t) = sint$, the velocity field represents the same curve pattern for all the involved system parameters.

**Author Contributions:** M.B.R.: Conceptualization, Investigation, Methodology, Software and final editing. J.A.: Data Curation, Data Anaysis, Project administration, Validation, Supervision, Formal analysis. A.U.R.: Validation, Investigation Writing, Reviewing and Editing, Initial writing, Visualization. All authors have read and agreed to the published version of the manuscript.

**Funding:** The work in this paper has been supported by the Polish National Science Centre under the grant OPUS 14 No. 2017/27/B/ST8/01330.

**Institutional Review Board Statement:** Not Applicable.

**Informed Consent Statement:** Not Applicable.

**Data Availability Statement:** During the current study, no data sets were developed or investigated. Thus, no data sharing is applicable to this article.

**Conflicts of Interest:** The authors professed that no conflict of interest for this publication, authorship and research of this article.

**Nomenclature**

| Symbol | Quantity | Units |
|---|---|---|
| $\omega$ | Non-dimensional velocity | $(-)$ |
| $\mu$ | Dynamic viscosity | $(\text{Kg} \cdot \text{m}^{-1} \cdot \text{s}^{-1})$ |
| $\theta$ | Dimensionless temperature | $(-)$ |
| $\upsilon$ | Kinematic coefficient of viscosity | $(\text{m}^2 \cdot \text{s}^{-1})$ |
| $Gr$ | Thermal Grashof number | $(-)$ |
| $g$ | Acceleration due to gravity | $(\text{m} \cdot \text{s}^{-2})$ |
| $T_w$ | Temperature of the plate | $(\text{K})$ |
| $\beta_T$ | Thermal expansion coefficient | $(\text{Kg} \cdot \text{m}^{-3})$ |
| $T_\infty$ | Temperature of fluid far away from the plat | $(\text{K})$ |
| $\rho$ | Fluid density | $(\text{Kg} \cdot \text{m}^{-3})$ |
| $\lambda_1$ | Relaxation time | $(-)$ |
| $\sigma$ | Electrical conductivity | $(\text{s} \cdot \text{m}^{-1})$ |
| $\lambda_2$ | Retardation time | $(-)$ |
| $C_p$ | Specific heat at constant pressure | $(\text{j} \cdot \text{Kg}^{-1} \cdot \text{K}^{-1})$ |
| $Pr$ | Prandtl number | $(-)$ |
| $s$ | Laplace parameter | $(-)$ |
| $M_0$ | Imposed magnetic field | $(\text{W} \cdot \text{m}^{-2})$ |
| $Q$ | Heat generation/absorption | $(\text{J} \cdot \text{K}^{-1} \cdot \text{m}^{-3} \cdot \text{s}^{-1})$ |
| $M$ | Total magnetic field | $(-)$ |
| $t$ | Time | $(\text{s})$ |
| $k$ | Thermal conductivity of the fluid | $(\text{W} \cdot \text{m}^{-2} \cdot \text{K}^{-1})$ |
| $P$ | Pressure | $(\text{N} \cdot \text{m}^{-2})$ |

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
