# Peer review of "Functional Effects of Permeability on Oldroyd-B Fluid under Magnetization: A Comparison of Slipping and Non-Slipping Solutions"

_applsci, doi:10.3390/app112311477_

Round 1

Reviewer 1 Report

Major and Minor Points:

  1. Include the novelty of choosing the fluid and its applications.
  2. Why the three dimensional analysis was not taken into consideration –Justify.
  3. Assumptions, Nomenclature, symbols and abbreviations can be included
  4. Explain the significance of choosing Pr, Gr to particular range and constraining the non-dimensional terms.
  5. Results and Discussion are more generic it can be elaborated.
  6. Validation of the exact numerical results should be included.

Overall, the above set points need to be considered before acceptance.

Author Response

The authors are grateful to the respected reviewer for their positive comments and suggestions.

REFEREE# 1

.Comment No. 1 Include the novelty of choosing the fluid and its applications.

Reply:  In the existing literature, solutions available which are computed by using the Laplace transformation technique, Durbin’s numerical Algorithm, Stehfest’s and Zakian’s numerical algorithms . Novelty of our work, we have computed the Exact Solution of the proposed problem with parametric analysis of system parameters.

Comment No. 2 Why the three dimensional analysis was not taken into consideration –Justify.

Reply: The unsteady laminar slip flow of an Oldroyd-B fluid together with heat transfer near an infinite vertical plate subjected to Newtonian heating is considered. The thermal radiation influence parallel to the plate is considered insignificant in contrast to that in the horizontal direction. It appears that any physical flow is generally three-dimensional. But these are difficult to calculate and call for as much simplification as possible. This is achieved by ignoring changes to flow in any of the directions, thus reducing the complexity, we have to consider two dimensional analysis.

Comment No. 3 Assumptions, Nomenclature, symbols and abbreviations can be included

 ‎ Reply: Nomenclature part with all used symbols and units are included in the updated version of the article.

Comment No. 4 Explain the significance of choosing Pr, Gr to particular range and constraining the non-dimensional terms.

Reply: The Prandtl number is a dimensionless quantity that puts the viscosity of a fluid in correlation with the thermal conductivity. It therefore assesses the relation between momentum transport and thermal transport capacity of a fluid. Fluids with small Prandtl numbers are free-flowing liquids with high thermal conductivity and are therefore a good choice for heat conducting liquids. Small values of the Prandtl number, Pr ≪ 1, means the thermal diffusivity dominates. Whereas with large values, Pr ≫ 1, the momentum diffusivity dominates the behavior

The Grashof number is a nondimensional number in fluid elements and warmth move which are the proportion of the lightness to thick power following up on a fluid. Grashof number is utilized when convective warmth move is occurring between the bodies.

The transition to turbulent flow occurs in the range 108 < Gr < 109 for natural convection from vertical flat plates. At higher Grashof numbers, the boundary layer is turbulent; at lower Grashof numbers, the boundary layer is laminar that is in the range 103 < Gr < 106.

Comment No. 5 Results and Discussion are more generic it can be elaborated.

Reply: The suggested correction has been made.

Comment No. 6 Validation of the exact numerical results should be included.

Reply: Done as per suggestion.

Reviewer 2 Report

Paper must be more carefully written. Also please check misprints, e.g. Prandtl, Grashof numbers, etc. instead of Pr there is Pr, and so on.

Author Response

The authors are grateful to the respected reviewer for his positive comments and suggestions.

REFEREE# 2

.

Comment No. 1 Paper must be more carefully written. Also please check misprints, e.g. Prandtl, Grashof numbers, etc. instead of Pr there is Pr, and so on.

Reply: The suggested changes have been made 

Reviewer 3 Report

  • There are inconsistent sentences (for example, the first sentence of the introduction), misprints in words (for example, "temprature", "oscilatroy" there) and obscure phrases (for example, "the role of temperature dissimilarity versus the temperature or time", "They suggested the finding and concluded that ... "and" the main theme of this manuscript is to have the significance "in the introduction). I think that the article needs to be worked out to improve the level of the English language.
  • The given image (Figure 1) does little to explain the geometry of the flow region. In addition, this image is shown before the first link to it.
  • There are no explanations for the notation introduced in the initial-boundary value problem (1) - (6) and expressions (7).
  • Sometimes it is not clear how to interpret the used notations. For example, "k" occurs in substitutions (7) and is deciphered after expressions (14). However, there is no certainty that the physical / mathematical meaning for the designation "k" is the same in both cases.
  • Extremely strange logic of a dividing the text into semantic parts: in section 3.1.1, only the designation of the Nusselt number is introduced. At the same time, section 3.1.2 is absent as such.
  • An Iillegal numbering system of variables is used: notations c_3 and c_4 are immediately introduced for the integration constants (see expression (21)) in the absence of notations c_1 and c_2. The situation is similar with the notations a_5, a_6, a_7 (there are no notations a_1,…, a_4) and other parameters. One gets the feeling that formulas and expressions are inserted into the article directly (by simple copying) from an preparatory calculation in a computer mathematics package without any processing of the results.
  • I consider the term "exact solution" inappropriate when using infinite series, Laplace transform and discrete convolution.
  • Phrases, which are used in the section "Results and discussion", do little to clarify the essence of the obtained results. For example, the phrase “The influence of the relevant dimensionless several connected parameters <…> are examined and portrayed graphically…” leaves open the question of what exactly these parameters affect.
  • In the comments to Figures 3-7, the word “slipping” was mentioned, which until then was found only in the title of the article. And accordingly, the difference in the mathematical model in the cases of the presence and absence of the slipping was not commented in any way.
  • "The outline layer of velocity profile gets thicker due to the fact that the small rate of thermal diffusion, Pr dominance the relative thickness of boundary layers of momentum in heat transfer problems." How can the dimensionless parameter (number Pr) and the layer thickness be compared?
  • In the final section "Conclusion" there are no conclusion about the effect of slipping on the process under study, announced in the title of the article. In other words, the results and discussions obtained have little to do with the stated goals.

Author Response

The authors are grateful to the respected reviewer for his positive comments and suggestions.

Comment No. 1 There are inconsistent sentences (for example, the first sentence of the introduction), misprints in words (for example, "temprature", "oscilatroy" there) and obscure phrases (for example, "the role of temperature dissimilarity versus the temperature or time", "They suggested the finding and concluded that ... "and" the main theme of this manuscript is to have the significance "in the introduction). I think that the article needs to be worked out to improve the level of the English language.

Reply: The suggested corrections have been made.

Comment No. 2 The given image (Figure 1) does little to explain the geometry of the flow region. In addition, this image is shown before the first link to it.

Reply: Improve the figure 1 as per suggestions and adjust the figure at appropriate place.

Comment No. 3 There are no explanations for the notation introduced in the initial-boundary value problem (1) - (6) and expressions (7).

 ‎ Reply: For explanation the notations which are used in the initial-boundary value problem (1) - (6) and expressions (7) are mentioned in the nomenclature part with all used symbols and units are included in the updated version of the article.

Comment No. 4 Sometimes it is not clear how to interpret the used notations. For example, "k" occurs in substitutions (7) and is deciphered after expressions (14). However, there is no certainty that the physical / mathematical meaning for the designation "k" is the same in both cases.

Reply: The suggested correction has been made. After expression (14) , a new constant  ‘q’ is used in place of the constant ‘k’ and also checked all the used notations in the problem carefully.

Comment No. 5 Extremely strange logic of a dividing the text into semantic parts: in section 3.1.1, only the designation of the Nusselt number is introduced. At the same time, section 3.1.2 is absent as such.

Reply: The suggested correction has been made. Expression for Nusselt number is included in the updated version of the article.

  • Comment No. 6 An Iillegal numbering system of variables is used: notations c_3 and c_4 are immediately introduced for the integration constants (see expression (21)) in the absence of notations c_1 and c_2. The situation is similar with the notations a_5, a_6, a_7 (there are no notations a_1,…, a_4) and other parameters. One gets the feeling that formulas and expressions are inserted into the article directly (by simple copying) from an preparatory calculation in a computer mathematics package without any processing of the results.

Reply: The suggested correction has been made. But it is necessary to mention that the constants  and  are not absent which are used in temperature expression (15), and  constants  and  are used in velocity expression (21). Also, the values of these constants are determined by using initial/ boundary conditions. Further, the constants    to        are mentioned after expression (24) and these constants are generated during the calculation of the velocity expression.

Comment No. 7 I consider the term "exact solution" inappropriate when using infinite series, Laplace transform and discrete convolution.

Reply:  We agree with the suggestion we used this term because most of the time in literature authors used exact solutions for such type of cases

Comment No. 8 Phrases, which are used in the section "Results and discussion", do little to clarify the essence of the obtained results. For example, the phrase “The influence of the relevant dimensionless several connected parameters <…> are examined and portrayed graphically…” leaves open the question of what exactly these parameters affect.

Reply: The suggested correction has been made. Detail for each parameter is given corresponding to figures in Result and discussion section.

Comment No. 9 In the comments to Figures 3-7, the word “slipping” was mentioned, which until then was found only in the title of the article. And accordingly, the difference in the mathematical model in the cases of the presence and absence of the slipping was not commented in any way.

Reply: It is clearly mentioned that in the Figures 3-7, velocity profile is plotted for the function   with and without slip effect γ.

  • Comment No.10 "The outline layer of velocity profile gets thicker due to the fact that the small rate of thermal diffusion, Pr dominance the relative thickness of boundary layers of momentum in heat transfer problems." How can the dimensionless parameter (number Pr) and the layer thickness be compared?

Reply: Boundary layer theory is used to describe the mechanism of heat transfer in fluids. In the heat transfer that occurs between a wall and a flowing fluid, heat is transported from the bulk of a fluid through a momentum boundary layer that consists of the bulk fluid and a transition layer and a thermal boundary layer that consists of stagnant film in which heat transport occurs by fluid conduction. The Prandtl number (Pr) of a fluid gives the relative importance of the momentum boundary layer to the thermal boundary layer in the transfer of heat. A high Pr number (> 5) means that heat transfer is more favorable to occur by fluid momentum than by thermal diffusion. In other words, high Pr number means that heat transfer is favored to occur by fluid momentum rather than by fluid conduction.

Comment No.11 In the final section "Conclusion" there are no conclusion about the effect of slipping on the process under study, announced in the title of the article. In other words, the results and discussions obtained have little to do with the stated goals.

.

Reply: The suggested correction has been made. In Result and discussion section the effects of slipping is discussed in detail, for all parameters we draw two parallel figures one represents the slipping effect and other figure shows without slipping effect. Also, include the remarks about the effect of slipping in the conclusion section.